# 20%-efficient polycrystalline Cd(Se,Te) thin-film solar cells with compositional gradient near the front junction

Deng-Bing Li [1], Sandip S. Bista [1], Rasha A. Awni [1], Sabin Neupane[1], Abasi Abudulimu[1], Xiaoming Wang[1], Kamala K. Subedi[1], Manoj K. Jamarkattel [1], Adam B. Phillips [1], Michael J. Heben [1], Jonathan D. Poplawsky [2], David A. Cullen [2], Randy J. Ellingson [1] & Yanfa Yan [1] ✉

Bandgap gradient is a proven approach for improving the open-circuit voltages ($V_{OC}$s) in Cu(In,Ga)Se$_2$ and Cu(Zn,Sn)Se$_2$ thin-film solar cells, but has not been realized in Cd(Se,Te) thin-film solar cells, a leading thin-film solar cell technology in the photovoltaic market. Here, we demonstrate the realization of a bandgap gradient in Cd(Se,Te) thin-film solar cells by introducing a Cd(O,S,-Se,Te) region with the same crystal structure of the absorber near the front junction. The formation of such a region is enabled by incorporating oxygenated CdS and CdSe layers. We show that the introduction of the bandgap gradient reduces the hole density in the front junction region and introduces a small spike in the band alignment between this and the absorber regions, effectively suppressing the nonradiative recombination therein and leading to improved $V_{OC}$s in Cd(Se,Te) solar cells using commercial SnO$_2$ buffers. A champion device achieves an efficiency of 20.03% with a $V_{OC}$ of 0.863 V.

Cadmium telluride (CdTe) photovoltaic is the most cost-competitive thin-film photovoltaic technology because of its low manufacturing cost and high module efficiency. Benefited from the reduced bandgap by the incorporation of selenium (Se) into CdTe to form a Cd(Se,Te) alloy absorber, the record power conversion efficiency (PCE) was improved from 16.5% in 2011 to over 22% in 2016[1,2]. The CdTe absorber has a bandgap of 1.54 eV and the Cd(Se,Te) absorber bandgap can be reduced to <1.4 eV due to the bandgap bowing effect, which is closer to the optimum bandgap value for single-junction solar cells per the Shockley-Queisser limit[3–5]. The lower bandgap accounts for the light absorption at longer wavelengths[6], attributing to the improvement of short-circuit current density ($J_{SC}$). It is also reported that grain boundary (GB) regions in Cd(Se,Te) films are more Se-rich than neighbouring grain interiors, suggesting that Se may passivate GBs[7].

Despite the increased $J_{SC}$, Cd(Se,Te) solar cells did not immediately realize significantly improved PCEs due to the compromised open-circuit voltage ($V_{OC}$)[3,8,9]. For a long time, only First Solar Inc, a leading Cd(Se,Te) module manufacturer, was able to fabricate Cd(Se,Te) solar cells showing PCEs significantly higher than that of CdTe solar cells[10]. It is only until recently when the commercial SnO$_2$ buffer layer was replaced by zinc magnesium oxide (ZMO), a significant progress was made in Cd(Se,Te) solar cells with PCEs up to 19% reported by several groups, and the ZMO/Cd(Se,Te)/CdTe configuration became the mainstream of CdTe solar cell research in the past years[11–13]. ZMO has a higher conduction band minimum (CBM) than Cd(Se,Te) alloy[14–16], therefore, a CBM spike is presented at the ZMO/Cd(Se,Te) interface and accumulates electrons therein, causing a downward band bending in the Cd(Se,Te) region and leading to an effective reduction of nonradiative recombination in the Cd(Se,Te) region near the front junction, which is critical for achieving high $V_{OC}$s[14,16]. This result suggests that the optoelectronic properties of the SnO$_2$/Cd(Se,Te) interface without the presence of a wide bandgap layer (e.g., ZMO) is not satisfactory and the nonradiative recombination near the front junction needs to be reduced. Though the ZMO buffer layer incorporation can amend this recombination, it has been rather difficult to reproducibly fabricate efficient Cd(Se,Te) solar cells

[1]Department of Physics and Astronomy, and Wright Center for Photovoltaics Innovation and Commercialization, University of Toledo, Toledo, OH 43606, USA. [2]Center for Nanophase Materials Sciences, Oak Ridge National Laboratory, Oak Ridge, TN 37831, USA. ✉e-mail: yanfa.yan@utoledo.edu

using ZMO as the buffer layer[17–19]. The major obstacle is the low electron conductivity of ZMO buffer, which is difficult to improve even with extrinsic doping[20,21]. When the ZMO layer, which forms a spike-like CBM alignment with Cd(Se,Te), is too resistive, the Cd(Se,Te) solar cells often show the so-called S-kink character in the photocurrent density-voltage (J–V) curves, which leads to low fill factors (FFs)[16,18,22]. Additionally, ZMO is sensitive to moisture, which may lead to long-term instability issues[23–25]. On the other hand, the commercial SnO₂ buffer is very stable and can be easily reproduced with desirable electron conductivity[26,27]. Benefit from these advantages, SnO₂ has been successfully used for decades in CdTe manufacturing[26–28]. Therefore, it is highly desirable to explore alternate approaches to reduce the recombination at the front interface to enable efficient Cd(Se,Te) solar cells fabrication using the commercial SnO₂ buffer.

Bandgap gradient is a proven approach for reducing the non-radiative recombination and improving the $V_{OC}$s in Cu(In,Ga)Se₂[29–32] and Cu(Zn,Sn)Se₂[33] solar cells. It is important to note that the bandgap gradient region must not present any additional detrimental hetero interface that often contains defects of nonradiative recombination. For this purpose, typically, homovalent elements were introduced to produce such a bandgap gradient. For example, varying the Ga/In ratio has long been used to introduce bandgap gradient in Cu(In,Ga)Se₂ to improve the $V_{OC}$s[29,30]. Recently, Ag was also incorporated in Cu(In,Ga)Se₂ and Cu(Zn,Sn)Se₂ to introduce regions with bandgap gradient[31,32]. For Cd(Se,Te) solar cells, a desirable approach for such a bandgap gradient is to incorporate a CdS thin layer in the front junction region. However, it has been previously found that an individual photo-inactive Cd(S,Se) layer with an additional detrimental interface (Supplementary Fig. 1b) would form after incorporating the CdS thin layer (Supplementary Fig. 1a), and such a structure led to even higher nonradiative recombination rates and lower $V_{OC}$s, $J_{SC}$, and PCEs[9,34].

Here, we report an approach to introduce a photoactive region with a desirable bandgap gradient without the formation of the detrimental interface. The key to this success was the incorporation of oxygenated CdS and CdSe layers (Cd(O,S), Cd(O,Se)) before the deposition of CdTe absorber layer. Upon CdCl₂ treatment, a region of penternary cadmium chalcogenide, Cd(O,S,Se,Te), was formed. This penternary cadmium chalcogenide has the same zinc blend structure of the absorber, without forming an additional detrimental hetero interface, and is photoactive. If pure CdS and CdSe layers were used, a photo-inactive wurtzite Cd(S,Se) layer would form, and consequently, an additional hetero interface would also form, consistent with the previous findings[9,34]. The Cd(O,S,Se,Te) region at the front interface has a wider bandgap than Cd(Se,Te) deeper in the absorber, introducing a desirable bandgap gradient that leads to a reduced hole density and a small CBM spike. SCAPS simulation revealed that this region exhibited significantly reduced nonradiative recombination and, therefore, improved $V_{OC}$s and PCEs for Cd(Se,Te) thin-film solar cells. The reduced nonradiative recombination was confirmed by time-resolved photoluminescence (TRPL) and photoluminescence quantum yield (PLQY) measurements. The successful introduction of the bandgap gradient region enabled efficient Cd(Se,Te) solar cell fabrication using a commercial SnO₂ buffer layer. All solar cells showed much improved $V_{OC}$s and PCEs compared with the solar cells without bandgap gradient. The champion Cd(Se,Te) solar cell achieved a PCE of 20.03%, with a $V_{OC}$ of 0.863 V, a $J_{SC}$ of 29.2 mA cm⁻², and a FF of 79.5%.

## Results and discussion
### Effects of bandgap gradient
We first discuss the beneficial effects of bandgap gradient in Cd(Se,Te) thin-film solar cells. Figure 1a shows the configuration of a Cd(Se,Te) thin-film solar cell using a commercial SnO₂ buffer layer as the n-type emitter. In the fabrication process, a CdTe layer is deposited on a CdSe layer as shown in Supplementary Fig. 1c. Upon CdCl₂ treatment, the CdSe and CdTe layers interdiffuse and form a Cd(Se,Te) absorber

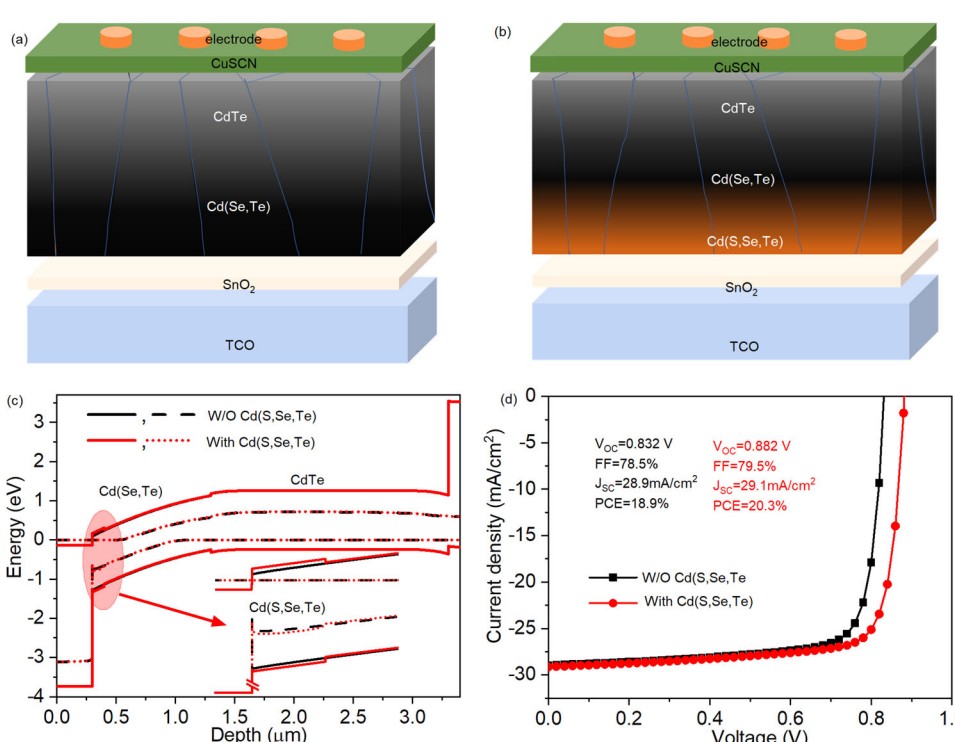

**Fig. 1 | Benefits of the bandgap gradient at the front junction.** Configurations of devices (**a**) without and (**b**) with a proposed bandgap gradient on the commercial SnO₂ buffer layer and (**c**) the corresponding band alignment of the proposed devices. The solid black and red lines are almost completely overlapped because of the small change of bandgap caused by the Cd(S,Se,Te) region. The inset is used to present the difference in the band alignment at the Cd(S,Se,Te) region. **d** SCAPS simulated J–V curves for devices without and with the bandgap gradient at the front interface.

region. From the front to the back junction, the Se concentration decreases and the Te concentration increases and the bandgap gradually increases from ~1.4 to ~1.5 eV (Supplementary Fig. 1d). However, Cd(Se,Te) thin-film solar cells with this configuration have not achieved high PCEs due to the relatively low $V_{OC}$s (the highest $V_{OC}$ reported in literature is 0.833 V). The low $V_{OC}$s in this configuration are probably due to the nonradiative recombination at the front interface, which can be reduced by introducing a wider bandgap region between the $SnO_2$ buffer and Cd(Se,Te) absorber. An intuitive approach to introduce such a region with a wider bandgap is to incorporate some S to form a Cd(S,Se,Te) quaternary compound at the front interface as shown in Fig. 1b. Due to the shallow p orbital of S, the bandgap of Cd(S,Se,Te) is expected to be wider than Cd(Se,Te) and introduces a small CBM spike. Different from the CBM spike at ZMO/Cd(Se,Te) interface, the spike is located inside the absorber region near the front interface. It does not introduce an additional interface and thereby reduces the recombination at the front interface. Additionally, the Cd(S,Se,Te) region should be less p-type than the Cd(Se,Te), indicating a lower hole density. Therefore, the S-incorporated region is expected to introduce a downward band bending in the absorber region near the front junction. Such a downward band bending would promote the separation of photo-generated electrons and holes, leading to significantly reduced recombination at the front junction region, which is beneficial for achieving high $V_{OC}$s.

1D SCAPS simulations were performed to confirm the benefit of this compositional and bandgap gradient. To simplify the simulation, the absorber consists of three regions, i.e., Cd(S,Se,Te), Cd(Se,Te), CdTe, without compositional gradient in each individual layer. The parameters of these regions are shown in Supplementary Table 1. We assume that these three regions have the same zinc blend structure and do not form detrimental hetero interfaces. We first considered a S concentration of ~10%, giving a bandgap of 1.5 eV. According to first principles calculations, the CBM and valence band maximum (VBM) of CdTe shift in reverse directions when alloyed with other chalcogenides[4,35]. Because of the low S concentration, a small CBM spike of 0.05 eV was used in the simulation. A free hole density of $2 \times 10^{14}$ cm$^{-3}$ in the 100 nm Cd(S,Se,Te) layer was used, which is slightly lower than the Cd(Se,Te) absorber layer ($5 \times 10^{14}$ cm$^{-3}$). The trap density at the $SnO_2$/absorber interface was set to be $6 \times 10^{13}$ cm$^{-3}$ to match the typical $V_{OC}$ reported for the device with the configuration of $SnO_2$/Cd(Se,Te)/CdTe[36].

The simulated band diagram is plotted in Fig. 1c. The CBM and VBM of the Cd(Se,Te) region show a clear downward band bending, which is beneficial for carrier separation. Since Cd(S,Se,Te) has a larger bandgap than Cd(Se,Te), a type I band alignment between Cd(S,Se,Te) and Cd(Se,Te) layers is formed with a 0.05 eV CBM offset (spike) and −0.07 eV VBM offset (inset in Fig. 1c) with a trend resembling the band alignment between ZMO and Cd(Se,Te)[16]. As demonstrated previously, such a band alignment can provide effective reduction on nonradiative recombination at the front interface[16,37]. The assumption of no additional detrimental interface suggests that bandgap gradient does not introduce Shockley-Read-Hall recombination ($J_{SRH}$). The simulation revealed that the recombination current density at the front interface ($J_{interf}$) is significantly reduced as compared with its counterpart without the Cd(S,Se,Te) region, especially when the forward bias is >0.7 V (Supplementary Fig. 2). The intersection of the total generated current density ($J_{gen}$) and total recombination current density, which is dominated by $J_{interf}$ in this work, determines the $V_{OC}$. Thus, the device with the compositional gradient has a lower $J_{interf}$ and a higher $V_{OC}$ (0.882 mV vs 0.832 mV) as shown in the simulated J–V curves. Thereby, the PCE increases to 20.3% from 18.9% (Fig. 1d). It is noted that, in both simulation configurations, the trap states at the $SnO_2$/absorber interface were fixed at the same concentration ($6 \times 10^{13}$ cm$^{-3}$). Therefore, the improved $V_{OC}$ is due to the reduced nonradiative recombination caused by the Cd(S,Se,Te) layer. We further simulated the effects of the

bandgap of the Cd(S,Se,Te) layer and trap states density at the front interface on solar cell performance (Supplementary Fig. 3). It is seen that increasing the bandgap of the Cd(S,Se,Te) layer and lowering the trap density at the front junction can lead to higher $V_{OC}$s.

## Approach to fabricate Cd(Se,Te) cells with Cd(S,Se,Te) region

The SCAPS simulation confirmed the benefit of including a Cd(S,Se,Te) region near the front interface. A natural strategy to fabricate Cd(Se,Te) solar cells with such a Cd(S,Se,Te) region is to deposit CdS, CdSe, and CdTe layers with desirable thickness as depicted in Supplementary Fig. 1a and intermix them through interdiffusion during subsequential $CdCl_2$ heat treatment. However, previous works have shown that this strategy does not work, because simply depositing these layers only leads to the formation of an individual wurtzite Cd(S,Se) layer (Supplementary Fig. 1b) with a hetero interface between with the Cd(Se,Te), as demonstrated by several groups[6,34,38–40]. First-principles calculations have shown that the mixing enthalpies of $CdS_{0.5}Se_{0.5}$, $CdSe_{0.5}Te_{0.5}$, $CdS_{0.5}Te_{0.5}$ are 3, 8, 25 meV, respectively, suggesting the high propensity to form Cd(S,Se)[4]. The Cd(S,Se) layer has a wurtzite crystalline structure and is typically photo-inactive, just like the CdS layer.

We found that the presence of oxygen during the deposition of the CdS and CdSe layers can effectively lower the crystallinity of CdS and CdSe, forming amorphous Cd(O,S) and Cd(O,Se) films (Supplementary Fig. 4), which have higher energies and are less stable than their polycrystalline counterparts. The higher energies enabled these layers to intermix with the CdTe layer to form a penternary Cd(O,S,Se,Te) region with a zinc blend structure near the front junction. If either of the CdS and CdSe layers was deposited without the presence of oxygen, the formation of the Cd(S,Se) inactive layer was inevitable. Additionally, the presence of oxygen can further increase the bandgap of Cd(S,Se,Te), which is beneficial for the reduction of the recombination at the front interface. The presence of the penternary Cd(O,S,Se,Te) region without the formation of hetero interface was confirmed by cross-sectional scanning transmission electron microscopy (STEM) (Fig. 2). The high-angle annular dark-field (HAADF) STEM images (Fig. 2a) showed clearly that there is an additional layer at the front interface in the stack with Cd(O,S)/pure CdSe layers (marked by the yellow arrow). Energy dispersive X-ray spectroscopy (EDS) mapping (Fig. 2b) and elemental line profiles (Fig. 2c) revealed that the additional thin layer is Cd(O,S,Se). However, there was no such layer observed in the stack with Cd(O,S)/Cd(O,Se) layers, and no clear interface was observed throughout the absorber layer (Fig. 2d). The elemental profiles (Fig. 2d and f) revealed that a penternary Cd(O,S,Se,Te) region with a compositional gradient formed near the front junction, implying that the Cd(O,S) and Cd(O,Se) layers fully intermixed with the CdTe layer and do not form any hetero interface. The absence of such an interface is a key to the bandgap gradient-induced improvement in $V_{OC}$s observed in the above SCAPS simulation. The comparison of the EDS mapping reveals that the presence of oxygen in the CdSe layer facilitates the complete intermixing of S with Se and Te.

The S compositional profile in the device with Cd(O,S)/Cd(O,Se) was characterized by time-of-flight secondary ion mass spectroscopy (TOF SIMS). The absorber layer was peeled off from the substrate and the TOF SIMS measurement was conducted from the Cd(O,S,Se,Te) side. O, S, Se, and Te profiles measured from TOF SIMS are shown in Supplementary Fig. 5. The initial concentrations of O, S, Se, and Te are $2.75 \times 10^{21}$ cm$^{-3}$, $9.88 \times 10^{20}$ cm$^{-3}$, $3.24 \times 10^{21}$ cm$^{-3}$, and $5.42 \times 10^{21}$ cm$^{-3}$, respectively, corresponding to a composition of $CdO_{0.222}S_{0.08}Se_{0.261}Te_{0.437}$. If O was not considered, the composition can be assumed to be $CdS_{0.102}Se_{0.336}Te_{0.562}$. To characterize the bandgap of this composition, we synthesized Cd(S,Se,Te) films with different S compositions by thermal co-evaporation. O was not

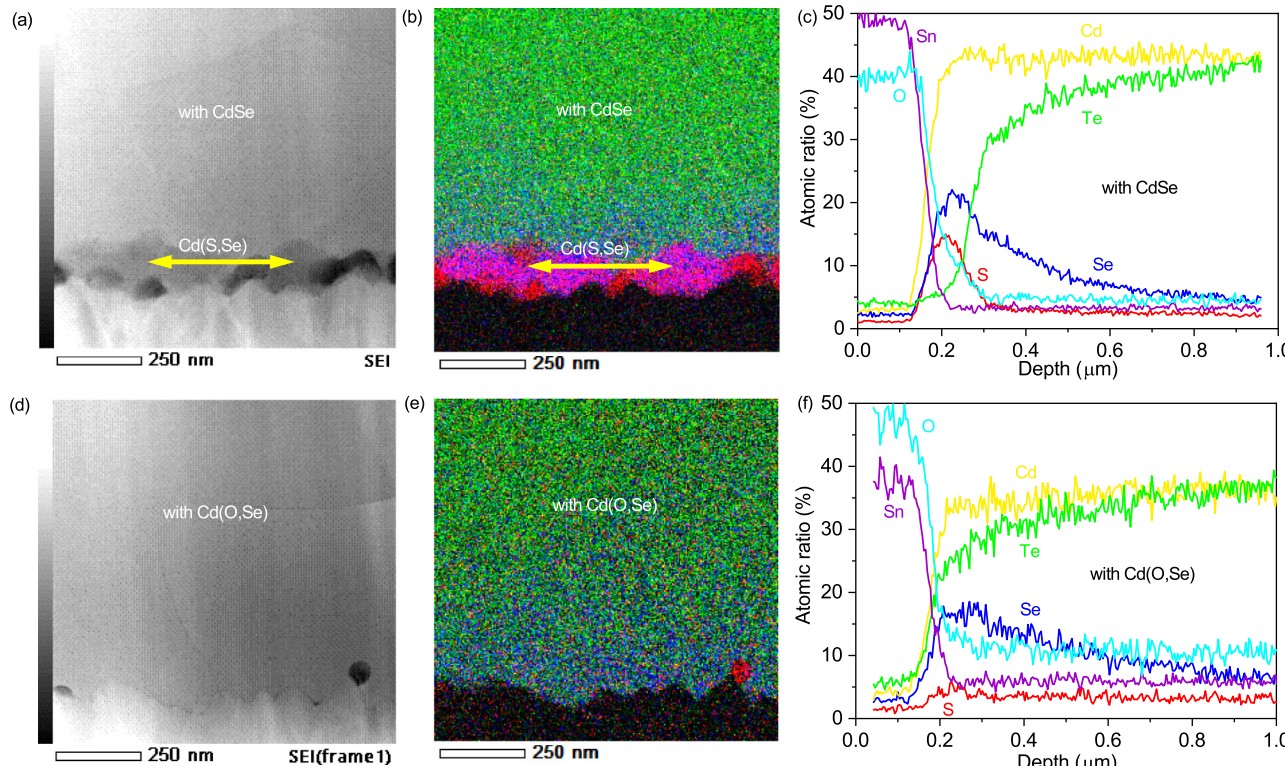

**Fig. 2 | Formation of a Cd(O,S,Se,Te) region in Cd(Se,Te) solar cells. a, d** Cross-sectional HAADF-STEM images; (**b, e**) overlaid EDS mappings for (green) Te, (blue) Se, and (red) S; (**c, f**) elemental line profiles extracted from the EDS maps for the devices with CdSe and Cd(O,Se).

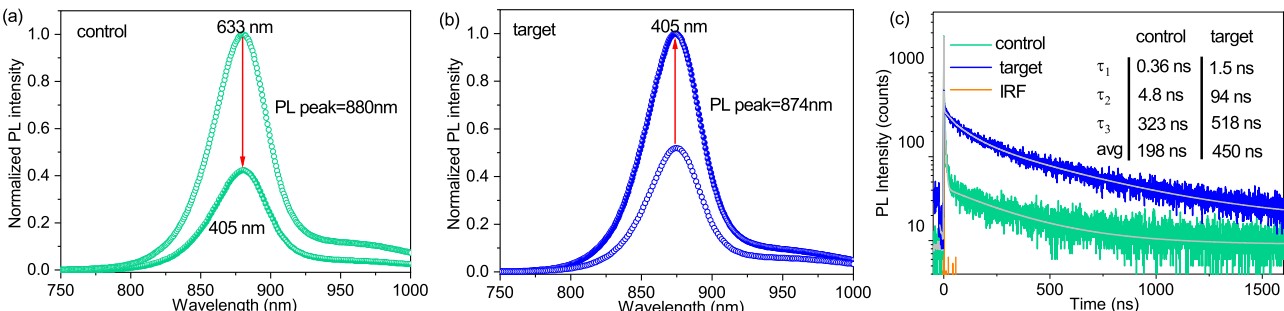

**Fig. 3 | Reduced recombination due to the presence of Cd(O,S,Se,Te) region. a** Steady-state PL spectra with different excitation beam wavelengths (405 and 633 nm) entering from the glass side for the (**a**) control and (**b**) target devices. **c** Time-resolved PL (TRPL) and the corresponding 3-exponential fitting curves for the control and target devices. The instrument response function (IRF) is also added as a reference. Note that TRPL was measured using a 633 nm excitation laser excited from the glass side.

incorporated due to the limitation of our evaporator. The bandgaps of Cd(S,Se,Te) films with various compositions were measured and depicted as a function of the S composition (Supplementary Fig. 6), from which the bandgap of $CdS_{0.102}Se_{0.336}Te_{0.562}$ was estimated to be about 1.5 eV. Therefore, with the incorporation of O, the bandgap of $CdO_{0.222}S_{0.08}Se_{0.261}Te_{0.437}$ should be >1.5 eV. The SIMS profile shows that the oxygen concentration in our solar cell is high at the interface and decreases rapidly until 0.5 μm into the absorber. Thus, toward the back contact from the front interface, the composition changes from Cd(O,S,Se,Te) to Cd(Se,Te) and then pure CdTe, the bandgap should correspondingly decreases from >1.5 eV to 1.38 eV and then increases back to 1.54 eV. Therefore, different from traditional CdTe solar cells with CdS as an n-type emitter, in this work, CdS provides a S source for diffusion into the absorber and introduce a bandgap gradient inside the absorber with commercial $SnO_2$ as n-type emitter instead.

## Verification of the recombination reduction

To confirm the effect of penternary Cd(O,S,Se,Te) region on device performance, a control device without S was also fabricated, which consists of a $SnO_2$/Cd(Se,Te)/CdTe stack (Supplementary Fig. 1c before $CdCl_2$ treatment and Supplementary Fig. 1d after $CdCl_2$ treatment). The steady-state photoluminescence (PL) spectra measurement with different excitation beam wavelengths (405 and 633 nm) entering from the glass side was conducted. The excitation beam with a longer wavelength penetrates farther into the absorber from the $SnO_2$/absorber interface. In the control device, the PL intensity decreases by ~60% when the excitation beam wavelength changes from 633 to 405 nm (Fig. 3a), suggesting pronounced nonradiative recombination at the $SnO_2$/Cd(Se,Te) interface compared to the bulk. While in the target device with a $SnO_2$/Cd(O,S,Se,Te)/Cd(Se,Te)/CdTe stack, the PL intensity with the 405 nm excitation beam is 200% of the PL intensity with the

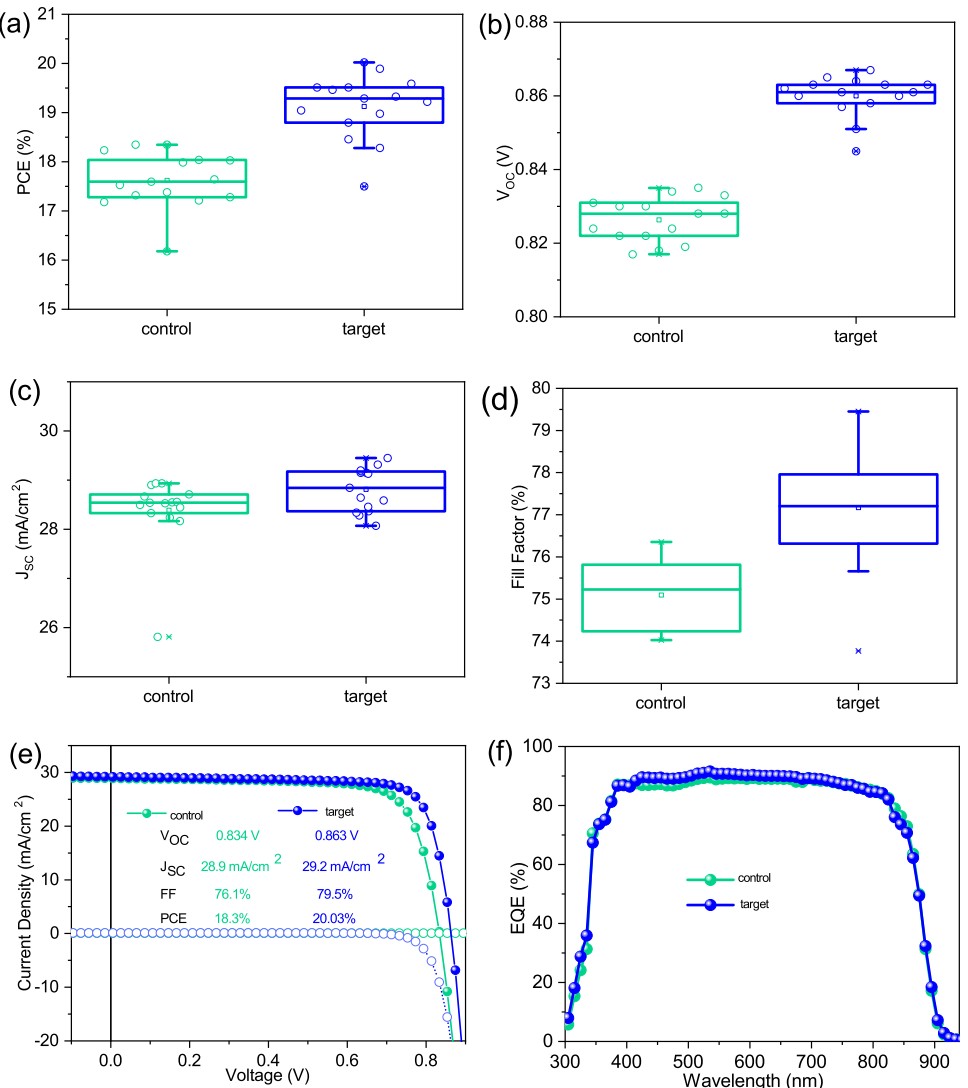

**Fig. 4 | Performance improvement of devices with the Cd(O,S,Se,Te) region.**
**a**–**d** The statistical distribution of the PV parameters (centre lines show the median, boxes show the upper and lower quartiles, and the whiskers show the full range; each black and red circle in the box and whisker plots presents the value of the *J*–*V* performances for one cell; cell number = 15) of (**a**) PCE, (**b**) $V_{OC}$, (**c**) $J_{SC}$ and (**d**) FF for the control and target devices. **e** *J*–*V* and (**f**) EQE curves for the champion control and target devices.

633 nm excitation beam (Fig. 3b), implies reduced nonradiative recombination at the front interface due to the Cd(O,S,Se,Te) region. The TRPL spectra, which were fitted using a 3-exponential decay fit, reveal that the target device shows a longer carrier lifetime ($\tau_1 = 1.5$ ns, $\tau_2 = 94$ ns, $\tau_3 = 518$ ns, $\tau_{avg} = 450$ ns) than the control device ($\tau_1 = 0.36$ ns, $\tau_2 = 4.8$ ns, $\tau_3 = 323$ ns, $\tau_{avg} = 198$ ns) (Fig. 3c). The much longer fast component of the TRPL lifetimes ($\tau_1$ and $\tau_2$) for the target sample suggests significantly reduced recombination at the front interface. The improvement of the slow component ($\tau_3$), which is commonly related to the bulk recombination, implies a longer survival time for the carriers in the bulk, enabling the carriers coming back to the absorber from the TCO in the later times and contributing to the TRPL decay[41]. PLQY was further used to predict the ideal $V_{OC}$ ($iV_{OC}$) and the $iV_{OC}$ difference ($\Delta iV_{OC}$) for the control and target devices[42,43]. The expected $\Delta iV_{OC}$ can be expressed as $\Delta iV_{OC} = (KT/q) \cdot \ln(PLQY_{target}/PLQY_{control})$. The target device delivers an $iV_{OC}$ of $893 \pm 12.2$ mV, which is ~35 mV higher than that of the control device ($858 \pm 13.5$ mV) (Supplementary Table 2). All these results indicate suppressed nonradiative recombination at the front interface in the device with a Cd(O,S,Se,Te) region.

## Characterization of the solar cell performance

The statistics of the performance parameters including the PCE, $V_{OC}$, $J_{SC}$, and FF of control and target devices are shown in Fig. 4a–d. Mature Cu doping was used for both the control and target devices[44,45]. It is seen that the control devices show $V_{OC}$s lower than 0.840 V, similar to that of Cd(Se,Te) solar cells without ZMO buffer layer previously reported by researchers[36]. On the other hand, all target cells show a clear improvement in $V_{OC}$s, FFs, and, therefore, PCEs. The $V_{OC}$ (0.863 V) of the champion device is as high as that reported for the champion Cd(Se,Te) solar cell using ZMO buffer layer[12,36,46]. It is noted that the target solar cells show significantly increased reproducibility as compared with Cd(Se,Te) solar cells using ZMO buffer layers (Supplementary Fig. 7), due to the reliable electric properties of commercial $SnO_2$ buffer layer. The *J*–*V* curves and the external quantum efficiency (EQE) curves of the champion control and target devices are shown in Fig. 3e, f, respectively. The champion target cell delivers a PCE of 20.03% with a $V_{OC}$ of 0.863 V, a $J_{SC}$ of 29.2 mA cm$^{-2}$, and a FF of 79.5%, while the best control solar cell shows a PCE of 18.3% with a $V_{OC}$ of 0.834 V, a $J_{SC}$ of 28.9 mA cm$^{-2}$, and a FF of 76.1%. The presence of the penternary Cd(O,S,Se,Te) region also slightly improves the quantum efficiencies at short wavelength region in the EQE curve (Fig. 3f).

We have demonstrated an approach to successfully introduce a bandgap gradient penternary Cd(O,S,Se,Te) region in Cd(Se,Te) solar cells without the formation of a photo-inactive layer and a detrimental hetero interface. This Cd(O,S,Se,Te) region is enabled by the incorporation of oxygenated CdS and CdSe layers. SCAPS simulation and TRPL and PLQY characterizations verified that the bandgap gradient penternary Cd(O,S,Se,Te) region led to reduced nonradiative recombination at the front interface. The introduction of such a bandgap gradient allowed the fabrication of efficient Cd(Se,Te) thin film solar cells using commercial $SnO_2$ buffer with a PCE of 20.03% and a $V_{OC}$ of 0.863 V.

## Methods

### Materials
CdS target (2-inch diameter, 99.99% purity, Plasma materials), CdS powder (99.999%, Materion), CdSe target (2-inch diameter, 99.99% purity, Plasma materials), CdSe powder (99.999%, Materion), CdTe chunks (99.999%, Materion), $CdCl_2$ (99.99%, Alfa Aesar), CuSCN (99%, Strem Chemical, Inc.), methanol (anhydrous, 99.8%, ACROS Organics), were commercial products. The solvent is of analytical purity grade and used as received without further purification.

### Device fabrication
The oxygenated CdS (60 nm) and CdSe layers (80 nm) were deposited using radio-frequency (RF) magnetic sputtering with a 2-inch target under 2% oxygen and 98% argon flow at room temperature according to our previous deposition method[47,48]. The bandgap of the optimum Cd(O,S) and Cd(O,Se) layers was characterized through UV-Vis measurement with a bandgap of 2.60 and 2.57 eV (Supplementary Figs. 8 and 9), respectively. CdTe absorber (3.5 μm) was then deposited by close-space sublimation (CSS) at the source and substrate temperatures of 660 and 590 °C, respectively, at 10 Torr, followed by a wet cadmium chloride ($CdCl_2$) treatment at 400 °C for 50 min in dry air. CuSCN (~50 nm) was deposited by the spin-coating method using 2 mg ml$^{-1}$ solution in 30 wt% ammonium hydroxide with 2000 rpm spin speed for 30 s. Then, a rapid thermal annealing process at 180 °C was used to facilitate the Cu diffusion according to our previous report[44,45]. After the rapid thermal annealing treatment, an Au layer (40 nm) was evaporated on the back surface with an individual device area of 0.08 cm$^2$. No further annealing treatment was taken after the Au deposition. Finally, a layer of 100 nm magnesium fluoride ($MgF_2$) anti-reflective was deposited for the champion devices on the glass side of the FTO substrate in an e-beam evaporation system.

### Characterization of the CdTe device
The Solar cell performance was characterized at room temperature in air by measuring current density-voltage ($J$–$V$) curves with a scanning speed of 200 mV s$^{-1}$ under AM1.5 G illumination using a solar simulator (PV Measurements Inc.) and a source meter (Keithley 2400). The light intensity for the $J$–$V$ measurements was calibrated by a standard silicon wafer solar cell certified by Newport. Before $J$–$V$ measurement, light soaking treatment was carried out for all the devices at 85 °C under AM1.5 G illumination for 15 min. A quantum efficiency system (model IVQE8-C, PV Measurements Inc.) was used to measure the quantum efficiency of the PSCs. A standard silicon wafer solar cell was used as the reference for the EQE measurement. The specimen foils for STEM were prepared using an FEI FIB dual-beam system using a standard in situ lift-out method. STEM imaging was performed using a JEOL NEOARM STEM equipped with dual 100 mm$^2$ windowless solid state device (SDD) spectrometers. STEM imaging was performed at 200 kV with a probe current of roughly 400 pA. Line scans were extracted from EDS maps using the JEOL Analysis Station software. The steady-state PL spectra were measured in a custom-built system, equipped with Horiba Symphony-II CCD detector and Horiba iHR320 monochromator, exciting the samples with 405 nm and 633 nm continuous

wave lasers ($2.5 \times 10^{17}$ cm$^{-2}$. S$^{-1}$). The Time-resolved PL measurements were done with Becker & Hickl TCSPC (HPM100-50, SPM130) based system equipped with Fianium supercontinuum laser exciting the sample at 633 nm ($10^{11}$ cm$^{-2}$. Pulse$^{-1}$). To assess the PLQY of the samples, we first measured the photoluminescence signal from the sample by exiting it with an optically chopped 405 nm laser and detecting the PL signal with a photodiode coupled with a lock-in amplifier (laser beam is filtered out before the photodiode using a long-pass filter). Then, we replaced the sample with a 5% Reflectance Standard (Labsphere Spectralon) and measured the diffused laser signal (no filter this time). Then, the signals measured from the sample and the Reflectance Standard are corrected for detector efficiency and also for the transmittance of the filter before calculating.

## Reporting summary
Further information on research design is available in the Nature Portfolio Reporting Summary linked to this article.

## Data availability
Source data are provided with this paper.

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

## Acknowledgements

This work is based on research sponsored by Air Force Research Laboratory under agreement number FA9453-18-2-0037 and FA9453-21-C-0056, National Science Foundation (contract No. 1665028 and 1711534), and by the U.S. DOE's Office of Energy Efficiency and Renewable Energy (EERE) under Solar Energy Technologies Office (SETO) Agreement DE-EE0008974. The U.S. Government is authorized to reproduce and distribute reprints for Governmental purposes notwithstanding any copyright notation thereon. The views expressed are those of the authors and do not reflect the official guidance or position of the United States Government, the Department of Defense or of the United States Air Force. The appearance of external hyperlinks does not constitute endorsement by the United States Department of Defense (DoD) of the linked websites, or the information, products, or services contained therein. The DoD does not exercise any editorial, security, or other control over the information you may find at these locations. Approved for public release; distribution is unlimited. Public Affairs release approval #AFRL-2022-1869. We thank Dr. David Strickler from Pilkington North America Inc. for supplying us with FTO coated substrates. STEM-EDS was supported by the Center for Nanophase Materials Sciences (CNMS), which is a US Department of Energy, Office of Science User Facility at Oak Ridge National Laboratory. The authors would like to thank James Burns for preparing the STEM samples. This manuscript has been authored by UT-Battelle, LLC under Contract No. DE-AC05-00OR22725 with the U.S. Department of Energy. The United States Government retains and the publisher, by accepting the article for publication, acknowledges that the United States Government retains a non-exclusive, paid-up, irrevocable, world-wide license to publish or reproduce the published form of this manuscript, or allow others to do so, for United States Government purposes. The Department of Energy will provide public access to these results of federally sponsored research in accordance with the DOE Public Access Plan (http://energy.gov/downloads/doe-public-access-plan).

## Author contributions

D.-B.L. and Y.Y. conceived the idea and designed the experiments. D.-B. L. deposited films, fabricated devices, and conducted the

characterizations. S.S.B. and S.N. deposited Cd(O,S) and Cd(O,Se) films. K.K.S., A.A., and R.J.E. contributed to TRPL measurement. X.W. and R.A.A. performed calculation of bandgaps. D.A.C. and J.D.P. carried out TEM measurements. M.J., A.P., and M.H. helped with the data analysis. D.-B.L. and Y.Y. wrote the initial draft and all authors contributed to the final manuscript. Y.Y. directed the project.

## Competing interests

The authors declare no competing interests.
