## [Peer Review File · Nature Communications]

20%-efficient Polycrystalline Cd(Se,Te) Thin-Film Solar Cells with Compositional Gradient near the Front JunctionEditorial Note: This manuscript has been previously reviewed at another journal that is not operating a transparent peer review scheme. This document only contains reviewer comments and rebuttal letters for versions considered at *Nature Communications*.

REVIEWER COMMENTS

Reviewer #1 (Remarks to the Author):

The authors addressed all the previous concerns. I do not have any further comments. The manuscript can be published as is.

Reviewer #5 (Remarks to the Author):

This paper presents novel results on band grading in CdSeTe solar cells using oxide interface layer between the CdS buffer and CdSeTe absorber layer. The device results are impressive and show significant improvement in the Voc and FF with this novel interface layer. Some further clarification would, however, improve the manuscript.

Lines 110 to 112, The reference to the Cd(S,Se,O,Te) introducing a "spike" similar to MZO is misleading. The "spike" created by the MZO is at the emitter/ absorber interface whereas in this work the "spike" is in the absorber (p-type region). This is an important distinction and should be drawn out in the discussion.

In Fig 2 the presence of O appears to have completely interdiffused the S - this should be commented on in the text as some remnant S in the interface region is part of the claim of this paper. The role of the CdS layer appears to be to provide a source of S for diffusion into the absorber and not to provide an n-type emitter layer. The SCAPS band diagram in Fig. 1(c) does not make it clear what the n-type layer is - I assume that this is n-type SnO₂. Again, clarification of this will emphasize the novelty and value of this paper in gaining a deeper understanding of the effect of band grading in the depletion region of the absorber.

Response Letter

Reviewer #1 (Remarks to the Author):

The authors addressed all the previous concerns. I do not have any further comments. The manuscript can be published as is.

Response: We are pleased to know that the reviewer is satisfied with our response and revision.

Reviewer #5 (Remarks to the Author):

This paper presents novel results on band grading in CdSeTe solar cells using oxide interface layer between the CdS buffer and CdSeTe absorber layer. The device results are impressive and show significant improvement in the Voc and FF with this novel interface layer. Some further clarification would, however, improve the manuscript.

Response: We thank the reviewer for the positive comments.

Lines 110 to 112, The reference to the Cd(S,Se,O,Te) introducing a "spike" similar to MZO is misleading. The "spike" created by the MZO is at the emitter/ absorber interface whereas in this work the "spike" is in the absorber (p-type region). This is an important distinction and should be drawn out in the discussion.

Response: We thanks the reviewer for the suggestion. In the revised manuscript, we made the following changes: Due to the shallow p orbital of S, the bandgap of Cd(S,Se,Te) is expected to be wider than Cd(Se,Te) and introduces a small CBM spike. Different from the CBM spike at ZMO/Cd(Se,Te) interface, this spike is located inside the absorber region near the front interface. It does not introduce an additional interface and reduces the recombination at the front interface.”

In Fig 2 the presence of O appears to have completely interdiffused the S - this should be commented on in the text as some remnant S in the interface region is part of the claim of this paper. The role of the CdS layer appears to be to provide a source of S for diffusion into the absorber and not to provide an n-type emitter layer. The SCAPS band diagram in Fig. 1(c) does not make it clear what the n-type layer is - I assume that this is n-type SnO₂. Again, clarification of this will emphasize the novelty and value of this paper in gaining a deeper understanding of the effect of band grading in the depletion region of the absorber.

Response: We thank the reviewer for these suggestions. In the revised manuscript, we made the following changes:

In page 5, we added: “Fig. 1a shows the configuration of a Cd(Se,Te) thin-film solar cell using a commercial SnO₂ buffer layer as the n-type emitter.”

In page 6, we added: “Different from the CBM spike at ZMO/Cd(Se,Te) interface, this spike is located inside the absorber region near the front interface. It does not introduce an additional interface and reduces the recombination at the front interface.”

In page 9 and 10, we added: “The comparison of the EDS mapping reveals that the presence of oxygen in the CdSe layer facilitates the complete intermixing of S with Se and Te.”

REVIEWERS' COMMENTS

Reviewer #5 (Remarks to the Author):

The minor edits address the concerns and help to improve clarity and impact of the paper. No further modifications required.